# Deep Leakage from Gradients

**Ligeng Zhu    Zhijian Liu    Song Han**
Massachusetts Institute of Technology
`{ligeng, zhijian, songhan}@mit.edu`

## Abstract

Exchanging gradients is a widely used method in modern multi-node machine learning system (*e.g.*, distributed training, collaborative learning). For a long time, people believed that gradients are safe to share: *i.e.*, the training data will not be leaked by gradients exchange. However, we show that it is possible to obtain the private training data from the publicly shared gradients. We name this leakage as *Deep Leakage from Gradient* and empirically validate the effectiveness on both computer vision and natural language processing tasks. Experimental results show that our attack is much stronger than previous approaches: the recovery is *pixel-wise* accurate for images and *token-wise* matching for texts. Thereby we want to raise people's awareness to rethink the gradient's safety. We also discuss several possible strategies to prevent such deep leakage. Without changes on training setting, the most effective defense method is gradient pruning.

## 1   Introduction

Distributed training becomes necessary to speedup training on large-scale datasets. In a distributed learning system, the computation is executed parally on each worker and synchronized via exchanging gradients (both parameter server [15, 23] and all-reduce [3, 30]). The distribution of computation naturally leads to the splitting of data: Each client has its own the training data and only communicates gradient during training (says the training set never leaves local machine). It allows to train a model using data from multiple sources without centralizing them. This scheme is named as collaborative learning and widely used when the training set contains private information [18, 20]. For example, multiple hospitals train a model jointly without sharing their patients' medical data [17, 26].

Distributed training and collaborative learning have been widely used in large scale machine learning tasks. However, **does the "gradient sharing" scheme protect the privacy of the training datasets of each participant?** In most scenarios, people assume that gradients are safe to share and will not expose the training data. Some recent studies show that gradients reveal some properties of the training data, for example, property classifier [27] (whether a sample with certain property is in the batch) and using generative adversarial networks to generate pictures that look similar to the training images [9, 13, 27]. Here we consider a more challenging case: can we **completely** steal the training data from gradients? Formally, given a machine learning model $F()$ and its weights $W$, if we have the gradients $\nabla w$ w.r.t a pair of input and label, can we obtain the training data reversely? Conventional wisdom suggests that the answer is no, but we show that this is actually possible.

In this work, we demonstrate *Deep Leakage from Gradients* (**DLG**): sharing the gradients can leak private training data. We present an optimization algorithm that can obtain both the training inputs and the labels in just few iterations. To perform the attack, we first randomly generate a pair of "dummy" inputs and labels and then perform the usual forward and backward. After deriving the dummy gradients from the dummy data, instead of optimizing model weights as in typical training, we optimize the dummy inputs and labels to minimize the distance between dummy gradients and real gradients (illustrated in Fig. 2). Matching the gradients makes the dummy data close to the

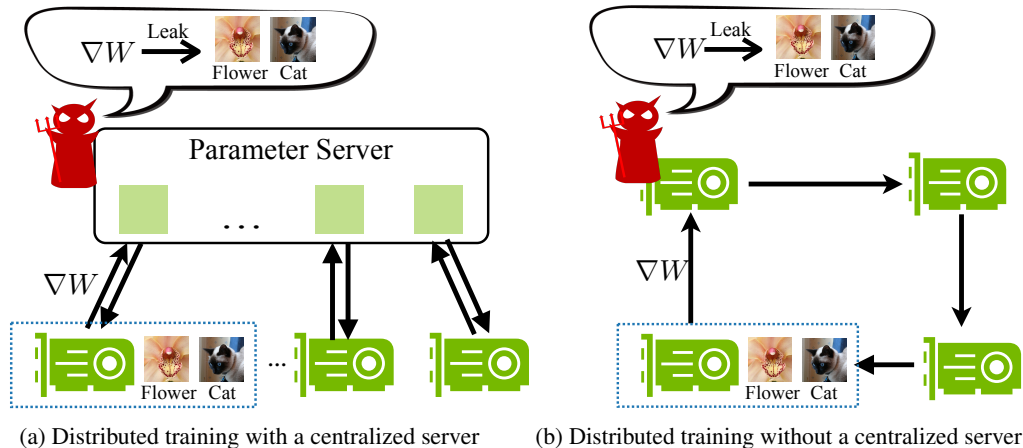

(a) Distributed training with a centralized server      (b) Distributed training without a centralized server

Figure 1: The deep leakage scenarios in two categories of classical multi-node training. The little red demon appears in the location where the deep leakage might happen. When performing centralized training, the parameter server is capable to steal all training data from gradients received from worker nodes. While training in a decentralized manner (e.g., ring all reduce [30]), any participant can be malicious and steal the training data from its neighbors.

original ones (Fig. 4). When the optimization finishes, the private training data (both inputs and labels) will be fully revealed.

Conventional "shallow" leakages (property inference [27, 34] and generative model [13] using class labels) requires extra label information and can only generate similar synthetic images. Our "deep" leakage is an optimization process and does not depend on any generative models; therefore, DLG does not require any other extra prior about the training set, instead, it can infer the label from shared gradients and the results produced by DLG (both images and texts) are the exact original training samples instead of synthetic look-alike alternatives. We evaluate the effectiveness of our algorithm on both vision (image classification) and language tasks (masked language model). On various datasets and tasks, DLG fully recovers the training data in just a few gradient steps. Such a deep leakage from gradients is first discovered and we want to raise people's awareness of rethinking the safety of gradients.

The deep leakage puts a severe challenge to the multi-node machine learning system. The fundamental gradient sharing scheme, as shown in our work, is not always reliable to protect the privacy of the training data. In centralized distributed training (Fig. 1a), the parameter server, which usually does not store any training data, is able to steal local training data of all participants. For decentralized distributed training (Fig. 1b), it becomes even worse since any participant can steal its neighbors' private training data. To prevent the deep leakage, we demonstrate three defense strategies: gradient perturbation, low precision, and gradient compression. For gradient perturbation, we find both Gaussian and Laplacian noise with a scale higher than $10^{-2}$ would be a good defense. While half precision fails to protect, gradient compression successfully defends the attack with the pruned gradient is more than 20%.

Our contributions include:

- We demonstrate that it is possible to obtain the private training data from the publicly shared gradients. To our best knowledge, DLG is the first algorithm achieving it.

- DLG only requires the gradients and can reveal pixel-wise accurate images and token-wise matching texts. While conventional approaches usually need extra information to attack and only produce partial properties or synthetic alternatives.

- To prevent potential leakage of important data, we analyze the attack difficulties in various settings and discuss several defense strategies against the attack.

## 2 Related Work

### 2.1 Distributed Training

Training large machine learning models (e.g., deep neural networks) is computationally intensive. In order to finish the training process in a reasonable time, many studies worked on distributed training to speedup. There are many works that aim to improve the scalability of distributed training, both at the algorithm level [7, 11, 16, 23, 32] and at the framework level [1, 2, 6, 29, 33]. Most of them adapt synchronous SGD as the backbone because the stable performance while scaling up.

In general, distributed training can be classified into two categories: with a parameter server (centralized) [15, 19, 23] and without a parameter server (decentralized) [3, 30, 33]). In both schemes, each node first performs the computation to update its local weights, and then sends gradients to other nodes. For the centralized mode, the gradients first get aggregated and then delivered back to each node. For decentralized mode, gradients are exchanged between neighboring nodes.

In many application scenarios, the training data is privacy-sensitive. For example, a patient's medical condition can not be shared across hospitals. To avoid the sensitive information being leaked, collaborative learning has recently emerged [17, 18, 26] where two or more participants can jointly train a model while the training dataset never leave each participants' local server. Only the gradients are shared across the network. This technique has been used to train models for medical treatments across multiple hospitals [18], analyze patient survival situations from various countries [17] and build predictive keyboards to improve typing experience [4, 20, 26].

### 2.2 "Shallow" Leakage from Gradients

Previous works have made some explorations on how to infer the information of training data from gradients. For some layers, the gradients already leak certain level of information. For example, the embedding layer in language tasks only produces gradients for words occurred in training data, which reveals what words have been used in other participant's training set [27]. But such leakage is "shallow": The leaked words is unordered and and it is hard to infer the original sentence due to ambiguity. Another case is fully connected layers, where observations of gradient updates can be used to infer output feature values. However, this cannot extend to convolutional layers because the size of the features is far larger than the size of weights.

Some recent works develop learning-based methods to infer properties of the batch. They show that a binary classifier trained on gradients is able to determine whether an exact data record (membership inference [27, 34]) or a data record with certain properties (property inference [27]) is included in the other participant's' batch. Furthermore, they train GAN models [10] to synthesis images look similar to training data from the gradient [9, 13, 27], but the attack is limited and only works when all class members look alike (e.g., face recognition).

## 3 Method

We show that it's possible to steal an image pixel-wise and steal a sentence token-wise from the gradients. We focus on the standard synchronous distributed training: At each step $t$, every node $i$ samples a minibatch $(\mathbf{x}_{t,i}, \mathbf{y}_{t,i})$ from its own dataset to compute the gradients

$$\nabla W_{t,i} = \frac{\partial \ell(F(\mathbf{x}_{t,i}, W_t), \mathbf{y}_{t,i})}{\partial W_t} \tag{1}$$

The gradients are averaged across the $N$ servers and then used to update the weights:

$$\overline{\nabla W_t} = \frac{1}{N} \sum_j^N \nabla W_{t,j}; \quad W_{t+1} = W_t - \eta \overline{\nabla W_t} \tag{2}$$

Given gradients $\nabla W_{t,k}$ received from other participant $k$, we aim to steal participant $k$'s training data $(\mathbf{x}_{t,k}, \mathbf{y}_{t,k})$. Note $F()$ and $W_t$ are shared by default for synchronized distributed optimization.

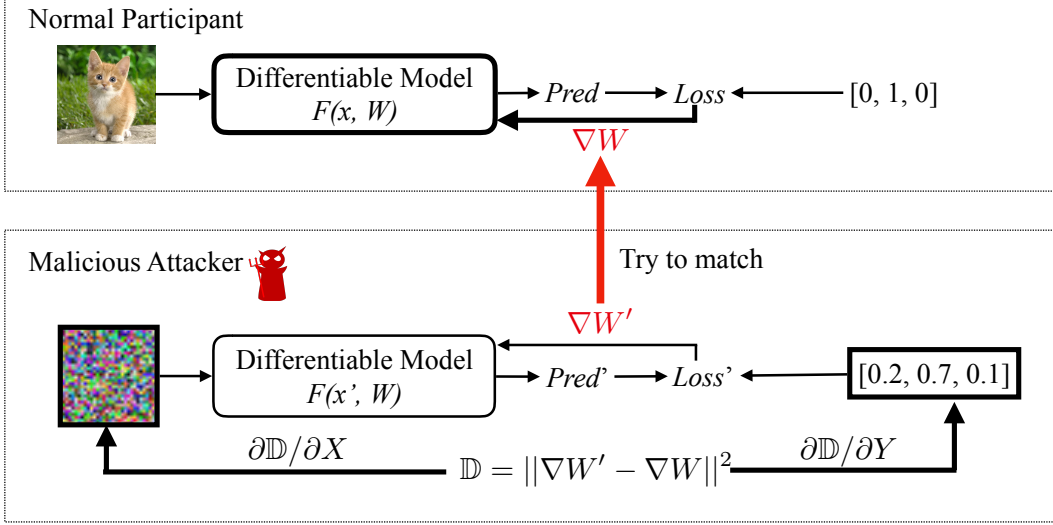

Figure 2: The overview of our DLG algorithm. Variables to be updated are marked with a bold border. While normal participants calculate $\nabla W$ to update parameter using its private training data, the malicious attacker updates its dummy inputs and labels to minimize the gradients distance. When the optimization finishes, the evil user is able to steal the training data from honest participants.

---

**Algorithm 1** Deep Leakage from Gradients.

---

**Input:** $F(\mathbf{x}; W)$: Differentiable machine learning model; $W$: parameter weights; $\nabla W$: gradients calculated by training data
**Output:** private training data $\mathbf{x}, \mathbf{y}$

1: **procedure** DLG($F, W, \nabla W$)
2:     $\mathbf{x}'_1 \leftarrow \mathcal{N}(0, 1) , \mathbf{y}'_1 \leftarrow \mathcal{N}(0, 1)$               ▷ Initialize dummy inputs and labels.
3:     **for** $i \leftarrow 1$ to $n$ **do**
4:         $\nabla W'_i \leftarrow \partial \ell(F(\mathbf{x}'_i, W_t), \mathbf{y}'_i)/\partial W_t$              ▷ Compute dummy gradients.
5:         $\mathbb{D}_i \leftarrow ||\nabla W'_i - \nabla W||^2$
6:         $\mathbf{x}'_{i+1} \leftarrow \mathbf{x}'_i - \eta \nabla_{\mathbf{x}'_i} \mathbb{D}_i , \mathbf{y}'_{i+1} \leftarrow \mathbf{y}'_i - \eta \nabla_{\mathbf{y}'_i} \mathbb{D}_i$    ▷ Update data to match gradients.
7:     **end for**
8:     **return** $\mathbf{x}'_{n+1}, \mathbf{y}'_{n+1}$
9: **end procedure**

---

To recover the data from gradients, we first randomly initialize a dummy input $\mathbf{x}'$ and label input $\mathbf{y}'$ (line 2 in Algo. 1). We then feed these "dummy data" into models and get "dummy gradients".

$$\nabla W' = \frac{\partial \ell(F(\mathbf{x}', W), \mathbf{y}')}{\partial W} \tag{3}$$

Optimizing the dummy gradients close as to original also makes the dummy data close to the real training data (the trends shown in Fig. 4). Given gradients at a certain step, we obtain the training data by minimizing the following objective

$$\mathbf{x}'^{*}, \mathbf{y}'^{*} = \underset{\mathbf{x}', \mathbf{y}'}{\arg\min} ||\nabla W' - \nabla W||^2 = \underset{\mathbf{x}', \mathbf{y}'}{\arg\min} ||\frac{\partial \ell(F(\mathbf{x}', W), \mathbf{y}')}{\partial W} - \nabla W||^2 \tag{4}$$

The distance $||\nabla W' - \nabla W||^2$ is differentiable w.r.t dummy inputs $\mathbf{x}'$ and labels $\mathbf{y}'$ can thus can be optimized using standard gradient-based methods. Note that this optimization requires $2^{nd}$ order derivatives. We make a mild assumption that $F$ is twice differentiable, which holds for the majority of modern machine learning models (e.g., most neural networks) and tasks.

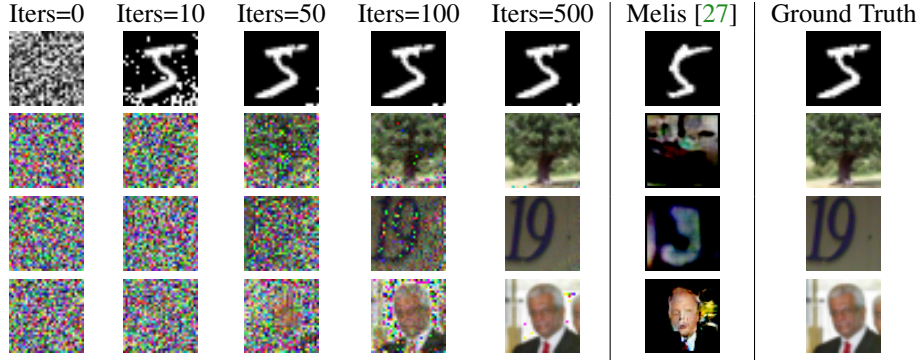

Figure 3: The visualization showing the deep leakage on images from MNIST [22], CIFAR-100 [21], SVHN [28] and LFW [14] respectively. Our algorithm fully recovers the four images while previous work only succeeds on simple images with clean backgrounds.

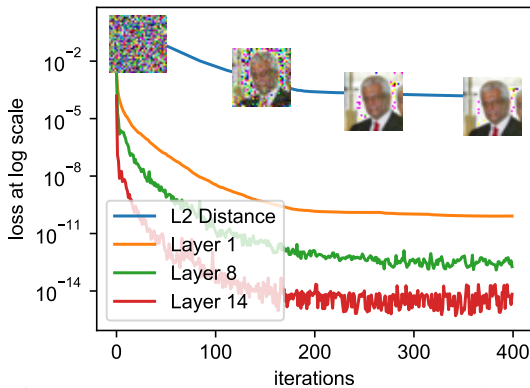

Figure 4: Layer-$i$ means MSE between real and dummy gradients of $i^{th}$ layer. When the gradients' distance gets smaller, the MSE between leaked image and the original image also gets smaller.

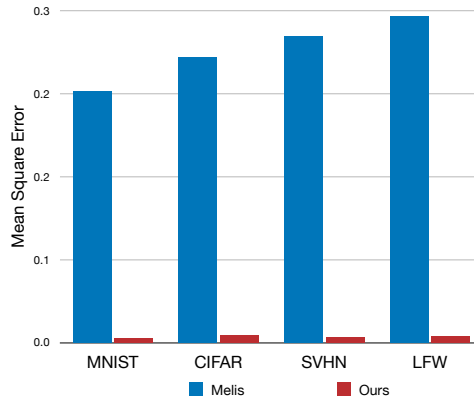

Figure 5: Compassion of the MSE of images leaked by different algorithms and the ground truth. Our method consistently outperforms previous approach by a large margin.

# 4 Experiments

**Setup.** Implementing algorithm. 1 requires to calculate the high order gradients and we choose PyTorch [29] as our experiment platform. We use L-BFGS [25] with learning rate 1, history size 100 and max iterations 20 and optimize for 1200 iterations and 100 iterations for image and text task respectively. We aim to match gradients from all trainable parameters. Notably, DLG has no requirements on the model's convergence status, in another word, *the attack can happen anytime during the training*. To be more general, all our experiments are using *randomly initialized weights*. More task-specific details can be found in the following sub-sections.

## 4.1 Deep Leakage on Image Classification

Given an image containing objects, images classification aims to determine the class of the item. We experiment our algorithm on modern CNN architectures ResNet-56 [12] and pictures from MNIST [22], CIFAR-100 [21], SVHN [28] and LFW [14]. Two changes we have made to the models are replacing activation ReLU to Sigmoid and removing strides, as our algorithm requires the model to be twice-differentiable. For image labels, instead of directly optimizing the discrete categorical values, we random initialize a vector with shape $N \times C$ where $N$ is the batch size and $C$ is the number of classes, and then take its softmax output as the one-hot label for optimization.

The leaking process is visualized in Fig. 3. We start with random Gaussian noise (first column) and try to match the gradients produced by the dummy data and real ones. As shown in Fig 4, minimizing the distance between gradients also reduces the gap between data. We observe that monochrome

| | Example 1 | Example 2 | Example 3 |
|---|---|---|---|
| Initial Sentence | tilting fill given **less word **itude fine **nton overheard living vegas **vac **vation *f forte **dis cerambycidae ellison **don yards marne **kali | toni **enting asbestos cutler km nail **oof **dation **ori righteous **xie lucan **hot **ery at **tle ordered pa **eit smashing proto | [MASK] **ry toppled **wled major relief dive displaced **lice [CLS] us apps _ **face **bet |
| Iters = 10 | tilting fill given **less full solicitor other ligue shrill living vegas rider treatment carry played sculptures lifelong ellison net yards marne **kali | toni **enting asbestos cutter km nail undefeated **dation hole righteous **xie lucan **hot **ery at **tle ordered pa **eit smashing proto | [MASK] **ry toppled identified major relief gin dive displaced **lice doll us apps _ **face space |
| Iters = 20 | registration , volunteer applications , at student travel application open the ; week of played ; child care will be glare . | we welcome proposals for tutor **ials on either core machine denver softly or topics of emerging importance for machine learning . | one **ry toppled hold major ritual ' dive annual conference days 1924 apps novelist dude space |
| Iters = 30 | registration , volunteer applications , and student travel application open the first week of september . child care will be available . | we welcome proposals for tutor **ials on either core machine learning topics or topics of emerging importance for machine learning . | we invite submissions for the thirty - third annual conference on neural information processing systems . |
| Original Text | Registration, volunteer applications, and student travel application open the first week of September. Child care will be available. | We welcome proposals for tutorials on either core machine learning topics or topics of emerging importance for machine learning. | We invite submissions for the Thirty-Third Annual Conference on Neural Information Processing Systems. |

Table 1: The progress of deep leakage on language tasks.

images with a clean background (MNIST) are easiest to recover, while complex images like face take more iterations to recover (Fig. 3). When the optimization finishes, the recover results are almost identical to ground truth images, despite few negligible artifact pixels.

We visually compare the results from other method [27] and ours in Fig. 3. The previous method uses GAN models when the class label is given and only works well on MNIST. The result on SVHN, though is still visually recognizable as digit "9", is no longer the original training image. The cases are even worse on LFW and collapse on CIFAR. We also make a numerical comparison by performing leaking and measuring the MSE on all dataset images in Fig. 5. Images are normalized to the range $[0, 1]$ and our algorithm appears much better results (ours $< 0.03$ v.s. previous $> 0.2$) on all four datasets.

## 4.2 Deep Leakage on Masked Language Model

For language task, we verify our algorithm on Masked Language Model (MLM) task. In each sequence, 15% of the words are replaced with a [MASK] token and MLM model attempts to predict the original value of the masked words from a given context. We choose BERT [8] as our backbone and adapt hyperparameters from the official implementation *.

Different from vision tasks where RGB inputs are continuous values, language models need to preprocess discrete words into embeddings. We apply DLG on embedding space and minimize the gradients distance between dummy embeddings and real ones. After optimization finishes, we derive original words by finding the closest entry in the embedding matrix reversely.

In Tab. 1, we exhibit the leaking history on three sentences selected from NeurIPS conference page. Similar to the vision task, we start with randomly initialized embedding: the reverse query results at iteration 0 is meaningless. During the optimization, the gradients produced by dummy embedding gradually match the original ones and so the embeddings. In later iterations, part of sequence

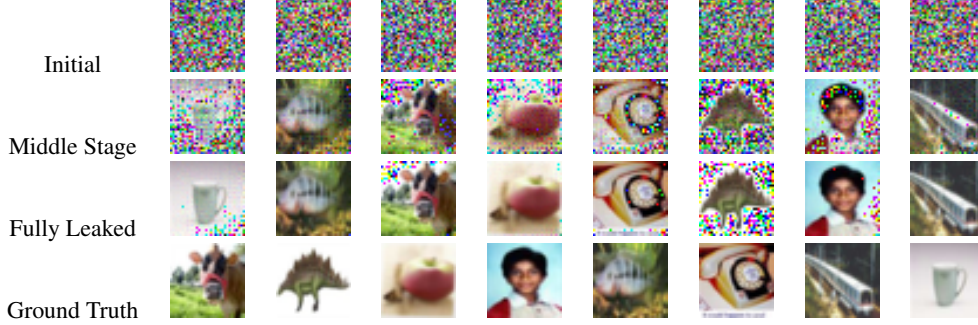

Figure 6: Results of deep leakage of batched data. Though the order may not be the same and there are more artifact pixels, DLG still produces images very close to the original ones.

gradually appears. In example 3, at iteration 20, 'annual conference' appeared and at iteration 30 and the leaked sentence is already close to the original one. When DLG finishes, though there are few mismatches caused by the ambiguity in tokenizing, the main content is already fully leaked.

### 4.3 Deep Leakage for Batched Data

The algo. 1 works well when there is only a single pair of input and label in the batch. However when we naively apply it to the case where batch size $N \geq 1$, the algorithm would be too slow to converge. We think the reason is that batched data can have $N!$ different permutations and thus make optimizer hard to choose gradient directions. To force the optimization closer to a solution, instead of updating the whole batch, we update a single training sample instead. We modify the *line 6* in algo. 1 to :

$$
\begin{aligned}
\mathbf{x}'^{i \bmod N}_{t+1} &\leftarrow \mathbf{x}'^{i \bmod N}_{t} - \nabla_{\mathbf{x}'^{i \bmod N}_{t+1}} \mathbb{D} \\
\mathbf{y}'^{i \bmod N}_{t+1} &\leftarrow \mathbf{y}'^{i \bmod N}_{t} - \nabla_{\mathbf{y}'^{i \bmod N}_{t+1}} \mathbb{D}
\end{aligned}
\tag{5}
$$

Then we can observe fast and stable convergence. We list the iterations required for convergence for different batch sizes in Tab. 2 and provide visualized results in Fig. 6. The larger the batch size is, the more iterations DLG requires to attack.

|  | BS=1 | BS=2 | BS=4 | BS=8 |
|---|---|---|---|---|
| ResNet-20 | 270 | 602 | 1173 | 2711 |

Table 2: The iterations required for restore batched data on CIFAR [21] dataset.

## 5 Defense Strategies

### 5.1 Noisy Gradients

One straightforward attempt to defense DLG is to add noise on gradients before sharing. To evaluate, we experiment Gaussian and Laplacian noise (widely used in differential privacy studies) distributions with variance range from $10^{-1}$ to $10^{-4}$ and central 0. From Fig. 7a and 7b, we observe that the defense effect mainly depends on the magnitude of distribution variance and less related to the noise types. When variance is at the scale of $10^{-4}$, the noisy gradients do not prevent the leak. For noise with variance $10^{-3}$, though with artifacts, the leakage can still be performed. Only when the variance is larger than $10^{-2}$ and the noise is starting to affect the accuracy, DLG will fail to execute and Laplacian tends to slight better at scale $10^{-3}$. However, noise with variance larger than $10^{-2}$ will degrade the accuracy significantly (Tab. 3).

Another common perturbation on gradients is half-precision, which was initially designed to save GPU memory footprints and also widely used to reduce communication bandwidth. We test two popular half-precision implementations IEEE float16 (*Single-precision floating-point format*) and

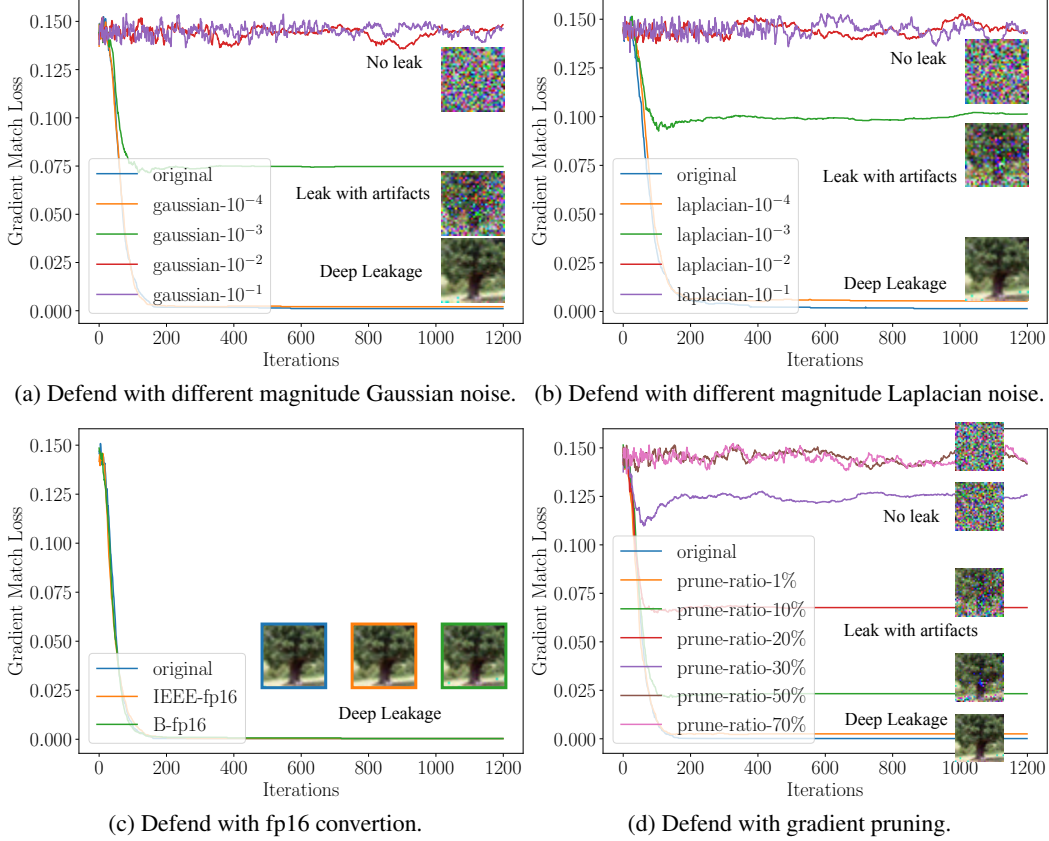

(a) Defend with different magnitude Gaussian noise.    (b) Defend with different magnitude Laplacian noise.

(c) Defend with fp16 convertion.    (d) Defend with gradient pruning.

Figure 7: The effectiveness of various defense strategies.

|  | Original | **G**-$10^{-4}$ | **G**-$10^{-3}$ | **G**-$10^{-2}$ | **G**-$10^{-1}$ | **FP**-16 |
|---|---|---|---|---|---|---|
| Accuracy | 76.3% | 75.6% | 73.3% | 45.3% | ≤1% | 76.1% |
| Defendability | – | ✗ | ✗ | ✓ | ✓ | ✗ |
|  |  | **L**-$10^{-4}$ | **L**-$10^{-3}$ | **L**-$10^{-2}$ | **L**-$10^{-1}$ | **Int**-8 |
| Accuracy | – | 75.6% | 73.4% | 46.2% | ≤1% | 53.7% |
| Defendability | – | ✗ | ✗ | ✓ | ✓ | ✓ |

Table 3: The trade-off between accuracy and defendability. **G**: Gaussian noise, **L**: Laplacian noise, **FP**: Floating number, **Int**: Integer quantization. ✓ means it successfully defends against DLG while ✗ means fails to defend (whether the results are visually recognizable). The accuracy is evaluated on CIFAR-100.

bfloat16 (*Brain Floating Point* [35], a truncated version of 32 bit float). Shown in Fig. 7c, both half-precision formats fail to protect the training data. We also test another popular low-bit representation Int-8. Though it successfully prevents leakage, the performance drops seriously (Tab. 3).

## 5.2 Gradient Compression and Sparsification

We next experimented to defend by gradient compression [24, 36]: Gradients with small magnitudes are pruned to zero. It's more difficult for DLG to match the gradients as the optimization targets are pruned. We evaluate how different levels of sparsities (range from 1% to 70%) defense the leakage. When sparsity is 1% to 10%, it has almost no effects against DLG. When prune ratio increases to 20%, as shown in Fig. 7d, there are obvious artifact pixels on the recover images. We notice that the maximum tolerance of sparsity is around 20%. When pruning ratio is larger, the recovered images are no longer visually recognizable and thus gradient compression successfully prevents the leakage.

Previous work [24, 36] show that gradients can be compressed by more than 300× without losing accuracy by error compensation techniques. In this case, the sparsity is above 99% and already

exceeds the maximum tolerance of DLG (which is around 20%). It suggests that compressing the gradients is a practical approach to avoid the deep leakage.

### 5.3 Large Batch, High Resolution and Cryptology

If changes in training settings are allowed, then there are more defense strategies. As suggested in Tab. 2, increasing the batch size makes the leakage more difficult because there are more variables to solve during optimization. Following the idea, upscaling the input images can also be a good defense, though some changes on CNN architectures is required. According to our experiments, DLG currently only works for a batch size up to 8 and image resolution up to 64×64.

Beside methods above, cryptology can also be used to prevent the leakage: Bonawitz *et al*. [5] designs a secure aggregation protocol and Phong *et al*. [31] proposes to encrypt the gradients before sending. Among all defenses, cryptology is the most secure one. However, both methods have their limitations and not general enough: secure aggregation [5] requires gradients to be integers thus not compatible with most CNNs, and homomorphic encryption [31] is against parameter server only.

## 6 Conclusions

In this paper, we introduce the *Deep Leakage from Gradients* (DLG): an algorithm that can obtain the local training data from public shared gradients. DLG does not rely on any generative model or extra prior about the data. Our experiments on vision and language tasks both demonstrate the critical risks of such deep leakage and show that such deep leakage can be only prevented when defense strategies start to degrade the accuracy. This sets a challenge to modern multi-node learning systems (e.g., distributed training, federated learning). We hope this work would raise people's awareness about the security of gradients and bring the community to rethink the safety of the existing gradient sharing scheme.

## Acknowledgments

We sincerely thank MIT-IBM Watson AI lab, Intel, Facebook and AWS for supporting this work. We sincerely thank John Cohn for the discussions.

## Footnotes

*https://github.com/google-research/bert

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
