[Supplementary Material]

# Supplementary Material for
# Deep Leakage from Gradients

## 1    Additional Vision Examples

We attach our visualization videos in following anonymized Google Drive https://drive.google.com/open?id=1u-NVQmXmHnPj57LYyZOxEbALUCgjhOPL.

Figure 1: The process of deep leakage on image classification task.

# 2   Additional Language Examples

| | Example 1 | Example 2 | Example 3 |
|---|---|---|---|
| Initial Sentence | daphne l **rud proceedings dishes **ps studio vatican **cious rafe gasp digitally **oir **oic news depth **ater rebel making cheese **gara = afl thee **sberg died inspirational elm **heart **ifice **ories indicates patch oclc **pressive gambia sim **jou dare va writ **minate **hot | spoiled laying **ching **ring tim **amen part phased **cus **rin gs ** **oue **sies **mity chance **sko ia **smith mirror odyssey **sman **1 envy icao interest rpm value taipei crambidae **ude rear **bha **cts joyah further [CLS] locomotive total itself yves intermediate inquiry | **dhi deux **cise state kenny walker **nent **iard **typical till **yne **quent **chua pair nad **N glint tara **kley en **chment roller traditionally **orf **vres **cellular impulse |
| Iters = 10 | daphne l friends proceedings dishes **ps studios vatican **cious rafe gasp digitally **oir **oic news depth **ater rebel making cheese **gara = afl thee **sberg died inspirational elm **heart **ifice trinity indicates patch oclc **pressive gambia sim **jou dare tore kept **minate dying | spoiled laying **ching **ring tim **amen part phased **cus **rin gs ** **oue **sies **mity chance engineer ia **smith mirror odyssey **sman **1 hosted icao interest rpm value taipei crambidae bedside rear **bha **cts joyah further [CLS] locomotive total itself yves intermediate inquiry | **dhi deux **cise state kenny walker **nent specialising **typical till **yne **quent **chua pair like **N glint tara fare en **chment memory traditionally solid mit **cellular impulse |
| Iters = 20 | daphne l friends proceedings dishes **ps studios vatican intersect **cious rafe gasp digitally **oir rotting news depth **ater rebel making cheese **gara = afl thee **sberg died inspirational elm **heart **ifice trinity indicates patch most **pressive gambia sim **jou dare tore kept **minate dying | spoiled laying **ching **ring tim **amen part phased **cus **rin gs ** **oue **sies **mity chance engineer ia **smith mirror odyssey **sman **1 hosted icao interest rpm value taipei crambidae bedside rear **bha **cts joyah further [CLS] locomotive total itself yves intermediate inquiry | **dhi deux apprentice state kenny walker trunk specialising **gs below **yne **quent fans pair like **N glint a wisdom en ku memory traditionally solid mit **cellular impulse |
| Iters = 30 | daphne l friends proceedings dishes **ps studios vatican specimen **cious rafe gasp digitally keeping rotting news depth coordination rebel making cheese **gara = carolina afl thee crop died inspirational elm **heart **ifice trinity indicates patch ( **pressive gambia hub **jou competitive tore kept **minate dying | spoiled laying nash **ring tim **ane part phased commercial **rin sad ** prayers **sies **mity chance line ia burden mirror odyssey **sman **1 hosted [MASK] arrival pact value recordings crambidae ll style **bha shop aria further himself never total itself adverse intermediate inquiry | highly ne **uri **ps papers walker trunk recurring its too ) included fans mix like algorithm glint a wisdom en ku memory traditionally position weapons varying impulse |
| Iters = 40 | submissions that violate the ne **uri **ps style or page limits , are not within the scope of ne **uri **ps ( see subject areas above ) , are in submission elsewhere , or have already been published elsewhere may be rejected without further review | spoiled during finally memorials ( confirmed ) flaws commercial by " kurdish **R including incorrect proof **s independent burden or disagreements 1966 - spaces they arrival , or software recordings crambidae should be amsterdam on as basis has without taking into consideration intermediate criteria | typical ne **uri **ps papers often ( but not always ) include a mix of algorithm **ic , theoretical , and experimental results , in varying proportions |
| Iters = 50 | submissions that violate the ne **uri **ps style or page limits , are not within the scope of ne **uri **ps ( see subject areas above ) , are in submission elsewhere , or have already been published elsewhere may be rejected without further review | submissions that have fatal ( confirmed ) flaws revealed by the reviewers — including incorrect proof **s , flawed or insufficient wet - lab , hardware , or software experiments — may be rejected on that basis , without taking into consideration other criteria | typical ne **uri **ps papers often ( but not always ) include a mix of algorithm **ic , theoretical , and experimental results , in varying proportions |
| Original Sentence | Submissions that violate the NeurIPS style or page limits, are not within the scope of NeurIPS (see subject areas above), are in submission elsewhere, or have already been published elsewhere may be rejected without further review. | Submissions that have fatal (confirmed) flaws revealed by the reviewers—including incorrect proofs, flawed or insufficient wet-lab, hardware, or software experiments—may be rejected on that basis, without taking into consideration other criteria. | Typical NeurIPS papers often (but not always) include a mix of algorithmic, theoretical, and experimental results, in varying proportions. |

Table 1: The process of deep leakage on masked language model task.