[Reviews · NeurIPS 2019]

Reviewer 1



This paper is easy to read and well structured. It raises an important privacy issue in naive collaborate learning. The problem statement is well-formalized and solution is well explained. I categorized this paper among the ones with major novelty and high significance. The following is some the detailed comments: 1- Authors have this assumption that F is twice differentiable. Although they discuss this briefly at some point where they replace ReLU with Sigmoid, but I would like to see deeper discussion regarding this constraint. What are the other common scenarios where attacker needs to replace the network layers? 2- In noisy gradients defense strategy, what is the tradeoff between information leakage and accuracy loss? 3- The concept of iteration in their paper is sometimes unclear. For example at the experimental setups when they mention: " ... max iterations 20 and optimize for 1200 iterations and 100 iterations for image and text 129 task respectively." When do you mean the number of iterations n in the for-loop in DLG algorithm? and when you mean it as the number of iterations in original distributed training? 4- does lower precision training helps? 8-bits precision training for example? do they protect training data? 5- DLG algorithm, line 4, computes the dummy gradient but it shows it as \Delta{W_t}. Should it be \Delta{W^'_t}? 6- Section 3.2: "to chooses" -> "to choose"

Reviewer 2



strength: a. Relevant findings - Given the recent surging interest in federate/collaborative learning, the authors' findings indicate that gradients do capture private information is insightful and relevant b. Elegant approach - The approach, unlike [27] is much simpler and requires weaker assumptions to reconstruct the input data. Major concerns: a. Attack model / gradient computation - The authors look at the specific case of reconstructing raw private inputs resulting from gradients resulting from a single iteration computed on a small batch of images. The attack model assumes these are shared to the adversary. - However, participants in collaborative/federated learning scenarios share gradients/updates computed over multiple batches and epochs (see [26], Alg. 1) -- after all, this is communication efficient. In this particular case, I'm skeptical of the effectiveness of the proposed attack. - Consequently, I'm concerned that the attack model (where attacker uses a single gradient) is in a contrived setting. - Moreover, while it could be argued that in some distributed computation models e.g., [14, 19, 23] the gradients from each iteration are indeed communicated -- these models seem to cater towards distributed computation in a cluster, in which the raw data is already possibly present for the adversary to access bypassing the need to use gradients. b. Missing details / writing I strongly recommend the authors to make more passes to fix typos/gramma and add many missing details that makes the findings unclear: - Implementation: * L135: CIFAR = CIFAR10 or CIFAR100? * L138: what is the batch size $N$ used in the experiments? * L134 / Eq. 4: Do you use all trainable parameters of the Resnet as $\nabla W'$? * L134: How were these models trained? What are their train/test accuracies? - Results: * Figure 5: Is the blue line "L2 distance" over all parameters and other lines over parameters of specific layers? Assuming it is, the green and red lines (parameters of layers closer to FC) have lower losses -- so why not use these? How do the leaked images look in this case? * Figure 3, 4, ..: are qualitative results from a held-out test set that was not used to train $W$? * Figure 3, 4, ..: How/why were these images chosen? How does the reconstruction look like on set of randomly sampled gradients? * Figure 7: what are the accuracies of the model when defending with these strategies? Afterall, if the accuracy of $W$ is retained with a prune ratio of 30% (Fig 7d), the attack can be easily defended. - Some unclear statements: * L119: "... batched data can have many different permutations ... N! satisfactory solutions ..." - How? Won't they still produce the same loss irrespective of the permutation? Other concerns: c. Experimental depth I overall find the experimental section somewhat shallow, leaving many questions unanswered: - Is the model $W$ trained to convergence? Isn't it more interesting to evaluate the attack at various stages of training of $W$? After all, the proposed attack seems relevant primarily at train-time. - How do the reconstruction results vary with batch size? - How does size/complexity of $W$ affect effectiveness of the attack? d. Other simple defenses - Given that "The deep leakage becomes harder when batch size increases." [Table 1], wouldn't this also make for a good defense? - Extending this argument and connecting to the point I raised earlier in (a): wouldn't averaging updates/gradients (computed over multiple batches) instead of gradients on a single batch also prevent reconstruction to a large extent? After all, the former is what's done in federated learning.

Reviewer 3



Good: It was surprising that obtaining the training datasets is possible by only utilizing the gradients in a collaborative learning scenario. And, the paper is easy to read and understand. Drawbacks or ‘should be improved’ Major 1. Since the proposed method utilizes the second order computation, L-BFGS, it will take quite a long time for computation (reconstruction). There is no information about it in the paper. 2. I think that comparisons with conventional attack methods (leakage) should be given in order to prove that your work is empirically better and the conventional approaches need extra information to obtain the leakage. 3. It is experimentally proven that the method is useful under a collaborative learning scenario. However, I think that the method itself is too simple. It is common to add noise to gradients or share fewer gradients in collaborative learning for defense or due to speed. So, a method that is more robust to noise injection or sparse gradient is more realistic. Minor 1. There are too many blank spaces in the paper as a whole. I think more contents such as discussion, figures or additional experiments can be added in the paper. 2. In figure 3, I think it is visually better just matching the order of generated and ground truth images. 3. I think that the paper will be more understandable if the dimension of given variables is written with symbols. 4. More explanations about algorithm 1 are needed in the writing.

[Author Response · NeurIPS 2019]

We thank all reviewers for their comments. All reviewers think it is an interesting paper. R1's review summarizes
our contribution well: "DLG is the first to shows a malicious player can recover private training data in collaborative
learning scenario." Both R1 and R3 are positive overall in their comments (R1 "easy to read and well structured",
"raises an important privacy issue", R3 "easy to read and understand", "it is surprising that obtaining the training datasets
is possible by only utilizing the gradients" ). For all typos/grammar mistakes, we have revised our writing accordingly.

**R2:** DLG may not work for accumulated gradients / Contrived settings.   This is a misunderstanding: DLG is still
effective in federated learning (Tab. 1). In the real-world case, a common workflow is to firstly deliver a pre-trained
model to users' devices and fine-tune it by Federated Learning[1]. In this case, the gradient and learning rate are both
small, thus the weight changes are small too. Thereby it can be approximated as multi-batch case and this is still
possible to attack. Nowadays, noisy / sparse / accumulated gradients are just *optional* choices for training acceleration,
but actually they are *essential* techniques to protect the training set. **Our work aims to raise people's awareness**
**about the security of gradients**.

| | Iterations=1 | Iterations=2 | Iterations=3 | Iterations=4 |
|---|---|---|---|---|
| MSE | $3.3 \times 10^{-6}$ | $3.5 \times 10^{-3}$ | $3.0 \times 10^{-3}$ | $1.8 \times 10^{-2}$ |

Table 1: The effectiveness of DLG on federated learning for different communication frequency.

| | Property Inference [26] | DLG |
|---|---|---|
| Eyeglasses | 0.94 | **1.00** |
| Asian | 0.92 | **1.00** |

Table 2: AUC score on LFW dataset.

14
**R3:** Comparison with previous work.   To the best of our acknowledge, **DLG is the first algorithm that performs**
**pixel-level and token-level leakage** based on shared gradients. We have compared conventional synthetic outputs and
our recovered results in the Fig. 4 in our paper. We also add a comparison on property inference task in Tab. 2: DLG is
significantly better since it can directly obtain the raw training data. In the revision, we will add more comparisons.

**R1, R2:** Trade-off between accuracy and defendability.   **R3:** The method is easy to defend.   **R1:** Does 8-bit help?
DLG is not easy to defend unless with a significant drop in accuracy. 8-bit gradient does not help either. We study
the trade-off between accuracy and defendability in Tab. 3. It shows that **only when the defense strategy starts to**
**degrade the accuracy then the deep leakage can be defended**.

| | Original | G-$10^{-4}$ | G-$10^{-3}$ | G-$10^{-2}$ | G-$10^{-1}$ | L-$10^{-4}$ | L-$10^{-3}$ | L-$10^{-2}$ | L-$10^{-1}$ | FP-16 | 8 bit |
|---|---|---|---|---|---|---|---|---|---|---|---|
| Accuracy | 76.3% | 75.6% | 73.3% | 45.3% | ≤1% | 75.6% | 73.4% | 46.2% | ≤1% | 76.1% | 53.7% |
| Defendability | – | ✗ | ✗ | ✓ | ✓ | ✗ | ✗ | ✓ | ✓ | ✗ | ✓ |

Table 3: **G**: Gaussian noise, **L**: Laplacian noise, **FP**: Floating number. ✓ means it successfully defends against DLG while ✗ means fails to defend. The accuracy is evaluated on CIFAR-100, same as what we used in the paper.

**R2:** Do you use all trainable parameters of the ResNet as $\nabla W$? Is the model $W$ trained to convergence?   Gradients
of *all* trainable parameters are used as $\nabla W$. It is important to clarify that DLG does *not* require the model trained to
converge: **The attack can be performed at any moment during the training** (Tab. 4). Our results in paper are based
on randomly initialized models.

| Train Progress | 0% epochs | 30% epochs | 70% epochs | 100% epochs |
|---|---|---|---|---|
| MSE | $5.7 \times 10^{-6}$ | $3.1 \times 10^{-7}$ | $4.4 \times 10^{-6}$ | $3.3 \times 10^{-6}$ |

Table 4: The MSE between leaked image and ground truth on different training stages. Pixel values are normalized to $[0, 1]$. The leaked image is nearly identical to original ones at each training phase.

**R1:** The concept of 'iterations'   refers to the "n" in the for-loop in DLG algorithm, not the training iterations.

**R2:** Fig 5. Is the blue line "L2 distance" over all parameters and other lines over parameters of specific layers?   No, the
L2 distance (on the top of the figure) is measured between the leaked image and original ground truth image. Other lines
are the distance between dummy gradients $\nabla W'$ and real gradients $\nabla W$ in each layer. We'll make it clear in the paper.

**R2:** Are qualitative results from a held-out test set? How/why were these images chosen?   Yes, qualitative results are
from a held-out test set. These images are randomly sampled and more examples have been already provided in the
appendix. There is no cherry-picking.

**R2:** DLG becomes harder (needs more iterations) to attack when batch size increases." Wouldn't this also make for a
good defense? How do the reconstruction results vary with batch size?   In multi-batch examples (Fig 3 in paper and
Line 1 in appendix), we only observe few artifact pixels compared with single-batch cases. Note DLG can be performed
off-line as long as current model status and gradients are saved. Large-batch is not a good defense strategy since the
information can be still leaked with more time and iterations.

**R3:** Time cost for L-BFGS reconstruction.   Though L-BFGS takes more calculations for every single step, it is still
faster (5 minutes) than other optimizers like SGD (30 minutes) on our hardware (Nvidia Tesla v100).

## Footnotes

[1]Towards federated learning at scale: System design `https://arxiv.org/abs/1902.01046`


[Meta-Review · NeurIPS 2019]

The paper presents an attack against federated learning algorithms and shows that when certain conditions apply, it may be possible to reconstruct the raw data from the gradients. This is an interesting observation. Federated learning, despite not having any formal privacy guarantees, is gaining popularity in corporates that operate on large amounts of data. In some cases, it might be used under the assumption that it provides privacy. Therefore, showing that this feeling is wrong may have real world impact. At the same time, the attack presented here may not be plausible in scenarios where gradients are batched. This is a borderline case, but I am leaning towards accepting since the results presented here may have real world impact on choices of large corporates.